# Thermal Boundaries in Cone Calorimetry Testing

**Sungwook Kang [1], Minjae Kwon [1], Joung Yoon Choi [1] and Sengkwan Choi [2,*]**

[1]  Fire Safety Centre, Korea Conformity Laboratories, Cheongju 28115, Korea; sungwookkang@kcl.re.kr (S.K.); mj.kwon@kcl.re.kr (M.K.); j.yoon.choi@kcl.re.kr (J.Y.C.)

[2]  Built Environment Research Institute, Ulster University, Newtownabbey BT37 0QB, UK

*  Correspondence: s.choi@ulster.ac.uk; Tel.: +44-28-9036-8300

**Abstract:** Bench-scale cone calorimetry is often used to evaluate the fire performance of intumescent-type coatings. During the tests, the coating geometry inflates. These thick, block-shaped specimens expose their perimeter side surfaces to both the heat source and the surroundings, unlike the typical thin, plate-shaped samples used in flammability tests. We assessed the thermal boundaries of block-shaped specimens using plain steel solids with several thicknesses. The heat transmitted through the exposed boundaries in convection and radiation modes was determined by four sub-defining functions: non-linear irradiance, convective loss, and radiant absorption into and radiant emission from solids. The individual functions were methodically derived and integrated into numerical calculations. The predictions were verified by physical measurements of the metals under different heating conditions. The results demonstrate that (1) considering absorptivity, being differentiated from emissivity, led to accurate predictions of time-temperature relationships for all stages from transient, through steady, and to cooling states; (2) the determined values for the geometric view factor and the fluid dynamic coefficient of convection can be generalized for engineering applications; (3) the proposed process provides a practical solution for the determination of optical radiative properties (absorptivity and emissivity) for use in engineering; and (4) the heat transmitted through the side surfaces of block specimens should be included in energy balance, particularly in the quantification of a heat loss mechanism. This paper outlines a comprehensive heat transfer model for cone calorimetry testing, providing insights into the mechanism of complex heat transmission generated on the test samples and quantifying their individual contributions.

**Keywords:** intumescent coating; cone calorimeter; thermal boundary condition; view factor; emissivity; absorptivity; convective heat transfer coefficient

## 1. Introduction

Bench-scale cone calorimetry has been widely accepted as a research tool for assessing the flammability of polymeric materials in the field of fire safety engineering. In the tests, a plate-shaped product sample is placed underneath the calorimeter and heated by a truncated cone-shaped heater, as shown in Figure 1a. In general, a pre-specified level of radiant heat is consistently incident upon, and uniformly dispersed over, the exposed square top surface of a given specimen during testing. The amount of the irradiance is initially measured at a designated position along the cone axis during the calibration stage (e.g., 25 mm under a conical heater), using a heat flux meter (Gardon or Schmidt–Boelter gauge). This conventional framework for flammability tests allows the application of a one-dimensional heat transfer mechanism along the *z*-axis.

In addition to the typical use of the instrument, it has often been adopted for examining the thermal performance of intumescent-type fire retardant polymers due to the well-controlled heating environment. When this type of material is exposed to the heater and reaches a critical temperature,

it decomposes and generates gaseous products during the intumescent stage. This thermochemical reaction leads to a significant volume increase during testing, as illustrated in Figure 1b, unlike the general case of material samples with stationary geometries (Figure 1a). The volume expansion normally increases the initial dry-film thickness (DFT) several tens of times. The surface area of the specimen's perimeter is simultaneously extended and considerably exposed to both the heat source and surroundings. Therefore, its deformational behaviour at elevated temperatures constructs a two-dimensional heat transfer environment along the *x* and *z* axes.

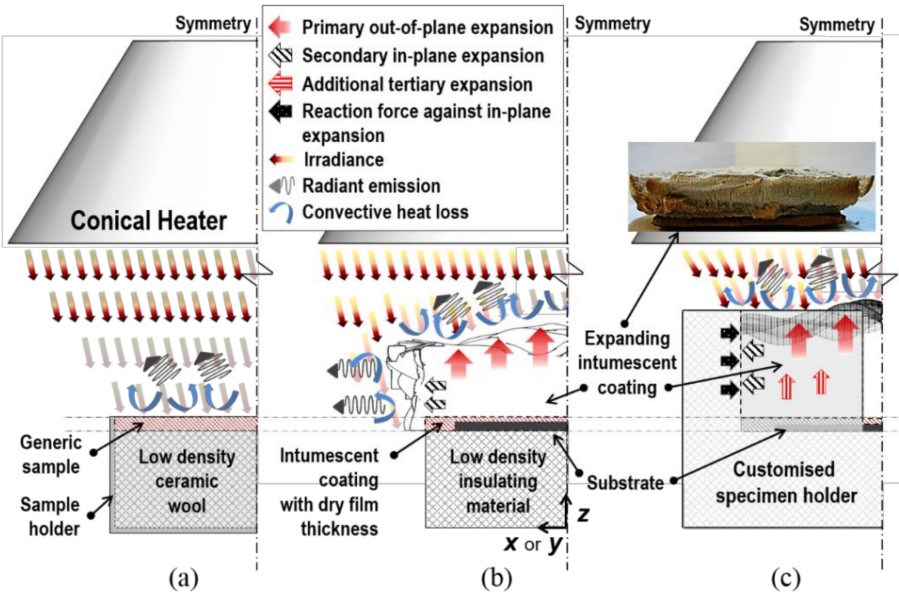

**Figure 1.** Schematics of thermal boundary conditions in cone calorimetry testing: (**a**) a typical material sample in flammability tests, (**b**) an intumescent-type specimen, and (**c**) an intumescent-type specimen with a customised holder.

With respect to the heat transmitted through the extended perimeter surfaces, developers of initial calorimeters considered how to manage the specimen edge condition and its effects on experimental results, particularly in the case of intumescing materials [1,2]. Although recommendations have been proposed, this issue is still unsolved. In other studies, the use of a customised specimen holder, which particularly envelops the sides, was suggested, as described in Figure 1c [3–6]. This test configuration was intended to employ the conventional, one-dimensional heat transfer model by neglecting the heat transfer occurring on the side surfaces. However, the holder structurally confines the substance with moving boundaries. It may provoke severe distortion or deformation during the course of intumescence, thus yielding unpredictable experimental results. This is because the volume expansion occurs not only toward the heater but also perpendicularly to the primary out-of-plane course. This secondary in-plane expansion is enforcedly restricted by the holder, which causes a tertiary out-of-plane displacement. The unconfined boundary condition shown in Figure 1b is, therefore, more suitable for testing this type of polymer to understand its actual thermal-structural behaviour.

This boundary condition has often been adopted in testing intumescent coatings [7–9]. The literature does not report the heat transfer occurring on side surfaces, since this was conventionally regarded as less significant than that generated on the top surface. However, the surface area of the four sides where the secondary heat transmission occurs exceeds half of the main top surface area (typically $100 \times 100$ mm$^2$) when the material object is thicker than 12.5 mm. Therefore, it is important to identify the contribution of the secondary heat transfer to the overall process to test specimens placed under the two-dimensional heat transfer environment. A clear interpretation of the derivations of parameters representing the heat transfer mechanism generated on sample boundaries is also lacking.

Hence, in this study, our aims were to (1) clarify the heat transfer mechanism, (2) quantify the components of the thermal boundaries, (3) verify the quantification, and (4) identify their contributions to the overall process when testing thick block-shaped samples instead of the typical thin plate-shaped specimens, under the unconfined boundary conditions illustrated in Figure 1b. This work was composed of the following functions:

- Non-uniform distribution of irradiance on both the exposed top and side surfaces due to the truncated cone-shaped heat source. The actual quantity of this distribution is theoretically predicted by calculating geometric view factors based on the contour integration method [10];
- Radiant absorption into the surfaces. The fraction of the irradiance that is actually absorbed by the areas is represented by absorptivity. This property is strongly dependent on optical aspects, such as the nature of absorbing surfaces and spectral/directional characteristics of incident radiation. In this study, a coupled numerical–experimental process was used for its determination;
- Heat losses from the surfaces by radiant emission. This radiation transfer is represented by emissivity. This radiative property is highly case-dependent on the nature of the materials and their surface conditions. In this study, radiant heat losses were determined using a cooling test procedure;
- Heat losses from the surfaces in convection mode, which are represented by the convective coefficient. Two sets of coefficients were individually defined for the dissimilar orientations of the top and side surfaces. Their derivations are demonstrated from correlations for buoyancy-induced air flows over horizontally and vertically-oriented planes, since such thermal energy transfers are correlated with physical fluid motions driven on the solids' boundaries.

In addition to the four sub-functions in regard to external thermal boundaries, heat penetration into a given specimen is an important issue for deriving a global conservation equation for energy because both the external and internal heat transmissions occur simultaneously in reality. In the case of block-shaped material objects, such as intumescent polymer and timber, heat penetrates through the media through a combination of conduction, convection, and radiation, creating an additional challenge in the engineering research field. Before examining these composite heat transfer mechanisms, the external thermal boundaries must be clarified, which was the primary intention of this research.

To verify the clarification and achieve this goal, the defined sub-functions were integrated into a series of numerical calculations constructed using the explicit finite difference method (FDM). The numerical models mainly predict the samples' nonlinear time–temperature relationship for all stages from transient, through steady, and to the cooling state. These predictions were compared with the results obtained from cone calorimetry tests. In the clarification and verification process in terms of thermal boundaries, plain steel blocks with different thicknesses (10, 15, and 20 mm) were adopted as block-shaped test samples. The application of a steel block specimen, instead of directly using an intumescent-type sample, benefits three aspects of the process: (1) The solid body preserves the boundaries in its existing state under severe heating conditions provided by the bench-scale apparatus. Thus, the time–temperature profiles can be precisely measured using thermocouples firmly welded to its boundary surfaces. (2) Its material properties allow reliable predictions with respect to the time–temperature relationship and (3) the metal block with high conductivity, a low Biot number, and a consistent internal structure at elevated temperatures allows the temperature gradients across its solid body to be disregarded. Drawing upon the results obtained from the verified numerical approach, individual contributions of the key functions to the overall heat transfer process can be demonstrated with a detailed analysis of the energy conserved in the material object.

To overcome difficulties in analysing a complex thermo-physical behaviour of an inorganic intumescent coating, we conducted a series of studies, as follows, and intensively examined the second: (1) experimental thermo-kinetic decomposition reactions of the coating [11]; (2) clarification of thermal boundaries of a thick, block-shaped specimen in cone calorimetry testing; (3) numerical analysis of internal heat transmission through a thick porous medium (i.e., fully expanded intumescent

coating) [12]; and (4) a numerical and experimental study of the integrated dynamic behaviour of the coating (i.e., intumescence) [13].

## 2. Heat Transfers in Cone Calorimeter Testing

### 2.1. External Heat Transfers

The thermal boundaries of a block specimen tested via cone calorimetry can be analysed by following a process that (1) classifies and specifies the heat transmission occurring during tests into multiple elements of heat transfer in different modes, and (2) quantifies each energy term by defining its controlling parameter. Figure 2 illustrates the detailed components in equation form with symbols Ⓐ–④, which can be categorised into three main processes: (1) heat gain via radiation transmission from the heater to the specimen (Ⓐ and Ⓑ), (2) heat loss via radiation emission by the sample (① and ③), and (3) heat loss in convection mode from the specimen (② and ④).

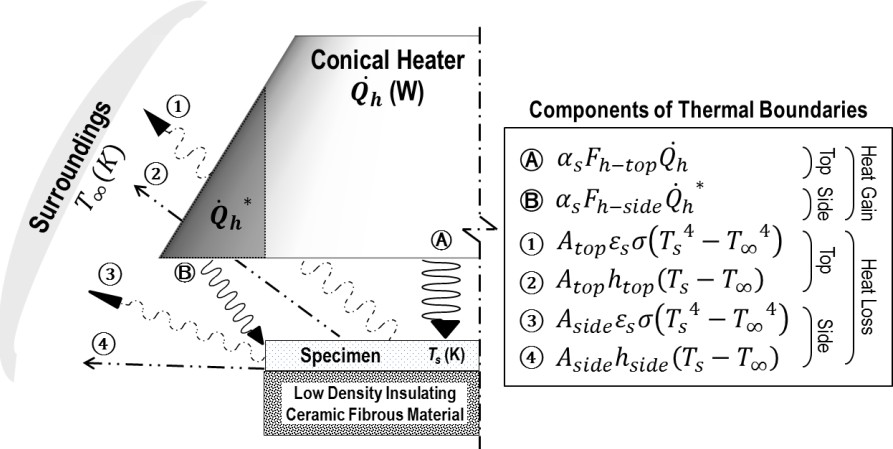

**Figure 2.** A schematic of the thermal boundaries of a specimen tested with the cone calorimeter.

### 2.1.1. Power of Heat Source

The radiant energy emitted by the conical heater per unit time ($\dot{Q}_h$ or $\dot{Q}_h^*$ in Watts) is an essential quantity when evaluating the radiation absorption into the specimen's exposed boundaries. However, this instrument does not directly supply the exact measurement of the radiant quantity; it only displays a mean temperature of the inlying spiral coil of the heater, which is thermostatically controlled, and the temperature data are automatically measured using built-in thermocouples held in contact with this heating element during tests. To estimate the quantity $\dot{Q}_h$, we adopted a theoretical approach that used (1) the reciprocity for the view factor ($F$) and (2) the irradiance data physically measured during calibration of irradiance using heat flux probes [10,14]. The approach starts from the derivation of an energy balance equation; the radiant energy that is incident on the measuring surface of a Gardon gauge ($\dot{Q}_g$) is equal to $\dot{Q}_h$ multiplied by the absorptivity of the probe and the view factor between the heater and the gauge, as follows:

$$\dot{Q}_g = \alpha_g F_{h-g} \dot{Q}_h \tag{1}$$

Once the two thermal quantities are expressed per unit area, Equation (1) becomes

$$A_g \dot{q}''_{abs,g} = \alpha_g F_{h-g} \left( A_h \dot{q}''_{emit,h} \right) \tag{2}$$

Hence, the radiant emittance of the heat source is

$$\dot{q}''_{emit,h} = \frac{\dot{q}''_{abs,g}}{\alpha_g F_{g-h}} \tag{3}$$

where $F_{g-h} = \frac{A_h}{A_g}F_{h-g}$, with respect to the radiation absorption into the perimeter side surfaces of the specimen, corresponding to energy term Ⓑ. These vertical areas are practically exposed to part of the total radiant flux of the heater (i.e., $\dot{Q}_h^*$), as illustrated with the shaded dark grey in Figure 2. This is due to the limited visibility as a result of the right-angular configuration between the heater and the side surfaces. This partial energy was quantified by calculating the ratio of the heater's total interior surface area to the partial surface area, which was viewed from the perimeter faces, based on the assumption that the interior surface was black and isothermal.

### 2.1.2. Radiation Absorption Mechanism (Ⓐ and Ⓑ)

The defined radiant power emitted by the heater ($\dot{Q}_h$ or $\dot{Q}_h^*$) is transported to the exposed surfaces of the specimen. In the course of the radiant intensity transport, the unique geometric configuration between the truncated, cone-shaped emitter and the block-shaped recipient causes non-uniform distributions of irradiance arriving over the top surface and the sides. The irradiance distribution is in direct relation to the spatial distances and angles created between diffuse infinitesimal area elements on the radiating surfaces of the two objects. The two geometric measures (distance and angle) vary according to the position of the elemental areas. This built-in nonlinearity of the cone calorimetry in respect to irradiance leads to apparent discrepancies when quantifying the net heat stored in the recipient at differential time intervals if a heat flux preset in the calibration stage is applied. The pre-specified heat flux setting is conventionally identified only at the centre of the sample's top surface. The issue in terms of the nonlinear irradiance was analytically solved using the geometric view factor. Its formulae and algebraic derivation were thoroughly scrutinised in our previous work [10]. Table 1 shows the parameter values calculated for the configuration between the standard bench-scale heater and the $100 \times 100 \times \delta$ mm$^3$ block specimens.

**Table 1.** View factors calculated for configuration between the standard-sized heater and steel plate.

| H [a] (mm) | Thickness of Specimens δ (mm) | View Factor | |
|:---:|:---:|:---:|:---:|
| | | Top Surface [b] ($F_{h\text{-}top}$) | Side Surface [c] ($F_{h\text{-}side}$) |
| 25 | 10 | 0.2508 | 0.0151 |
| 20 | 15 | 0.2627 | 0.0263 |
| 15 | 20 | 0.2730 | 0.0410 |

[a] The vertical distance between the frustum baseplate of the conical heater and the top surface of the sample tested; [b] for specimens with dimensions of $100 \times 100$ mm$^2$ (length × width); [c] values for one of the four side surfaces.

The radiant energy multiplied by the view factor is multiplied again by the absorptivity ($\alpha$) to quantify the actual radiant heat absorbed by the exposed boundaries. This radiative property has been conventionally considered as being equivalent to the emissivity ($\varepsilon$) in engineering applications based on Kirchhoff's law. This approximation is strictly true only in thermodynamic equilibrium, which indicates that no net heat transmission exists between two radiating objects [15]. However, in cone calorimetry tests, the heating process is practically a one-way enforced heat transport from the heater to the recipient in radiation mode, rather than a reciprocal heat exchange to achieve a thermodynamic equilibrium between the two objects. This heating environment indicates that the contribution of absorptivity is dominant over that of emissivity. In particular, the radiant absorption is much more significant than the radiant emission during the period of temperature increase. Hence, the independent use of the absorptivity, differentiated from the emissivity, is proposed to advance the quantification of the net heat stored in the sample tested with this instrument.

### 2.1.3. Radiation Emission Mechanism (① and ③)

The phenomenon of heat loss from the material object occurs mainly in radiation mode due to the high temperature, which is indicated by energy terms ① and ③. Unlike the mechanism of pure radiation

absorption, which is represented by the absorptivity (Ⓐ and Ⓑ), the absorption part of this radiant emission process (i.e., $\varepsilon_s \sigma T_\infty^4$) is represented by the emissivity ($\varepsilon$) based on Kirchhoff's law. In this case, the use of the approximation can be justified by two reasons: (1) the absorption part is incorporated in the radiant heat loss, which aims to reach a thermodynamic equilibrium between the heated sample ($T_s$) and the surrounding air at ambient temperatures ($T_\infty$), and (2) the impact of absorptivity is less significant than that of emissivity through the course of the heat transmission. Notably, the view factor is not involved in this mechanism as it is a non-directional diffuse radiant emission.

### 2.1.4. Convection Loss Mechanism

Convection transfer, caused by the generation of buoyancy-driven flows occurring adjacent to the exposed top and side surfaces of the heated samples, is also an important heat loss process during tests. From the fluid dynamics viewpoint, the anticipated fluid motion adjacent to the heated vertical side surfaces is different from that over the heated horizontal top surface. To describe dissimilar fluid motions, the application of differentiated convective coefficients ($h_{top}$ and $h_{side}$) is proposed to quantify convection losses ② and ④.

Due to the lack of attention paid to the heat transmitted through the vertical sides to date, little information was available on the coefficient $h_{side}$ in previous studies relevant to cone calorimetry tests. In reference to the convection transfer produced on the top surface, comprehensive attempts have been made to define $h_{top}$ using direct or indirect methods. Its derivation has been approached from three aspects: (1) the use of a standard correlation for convective motion over a plane [16,17]; (2) the formulation of an energy equilibrium model assigning the role of a single effective factor to the coefficient $h$, which represents convection transfers, as well as the uncertainties regarding the other heat transfer mechanisms developed [4,18–20]; and (3) the application of parameter values from previous studies [3,7,8,21,22].

In this research, we individually derived two coefficients (for convection transfers generated on the top and side surfaces) from convection correlations for buoyancy-driven flows over both horizontally and vertically-oriented planes, under uniform heating conditions, based on classical principles. The applicability of the correlations for the present cases was inferred from existing experimental data. Their reliability was examined through comparison with the coefficient values proposed in previous research [3,4,7,8,16–22].

### 2.2. Internal Heat Transfer

In addition to the external heat transfers conducted by the three main mechanisms introduced in Section 2.1, heat penetration into the plain steel blocks was examined. To efficiently predict the time-dependent temperature development of the material object with high conductivity, a lumped capacitance approach was used to derive the energy balance. This method is strictly applicable only when the temperature distribution throughout solids can be neglected. The Biot number ($Bi$) is the evaluation criterion of the degree of the temperature gradient. This number represents the ratio of internal conductive resistance ($R_{cond}$) to external convective and radiative resistances ($R_{conv\_out}$, $R_{rad\_in}$, and $R_{rad\_out}$) of systems, as described in Figure 3 and as follows:

$$Bi_{heat\_in} - Bi_{heat\_out} = \frac{R_{cond}}{R_{rad\_in}} - \frac{R_{cond}}{\left[1/R_{rad\_out} - 1/R_{conv\_out}\right]^{-1}} \leq 0.1 \tag{4}$$

where $R_{cond} = \frac{\delta}{A_s k}$, $R_{conv\_out} = \frac{1}{A_s h}$, $R_{rad\_in} = \frac{1}{A_s \alpha_s \left\{\varepsilon_h \sigma \left(T_h^2 + T_{s1}^2\right)(T_h + T_{s1})\right\}}$, $R_{rad\_out} = \frac{1}{A_s \varepsilon_s \sigma \left(T_{s1}^2 + T_\infty^2\right)(T_{s1} + T_\infty)}$. Since $Bi$ is proportional to the specimen's thickness ($\delta$), a 20-mm-thick metal sample was examined as a worst-case scenario. The maximum $Bi$ value for this solid was calculated as 0.0046, which is far less than 0.1. This indicates that the application of the lumped approximation is appropriate for all the steel blocks used. In the physical measurements (which are mainly introduced in Section 3), we also observed that the maximum temperature difference within the 20-mm-thick solid body was less than

10 K in steady state, which is approximately less than 1.8% of the recorded temperatures. Consequently, temperature distributions across the solids were negligible; therefore, we did not consider conduction transfers through the lumped volumes in the derivation of energy balance.

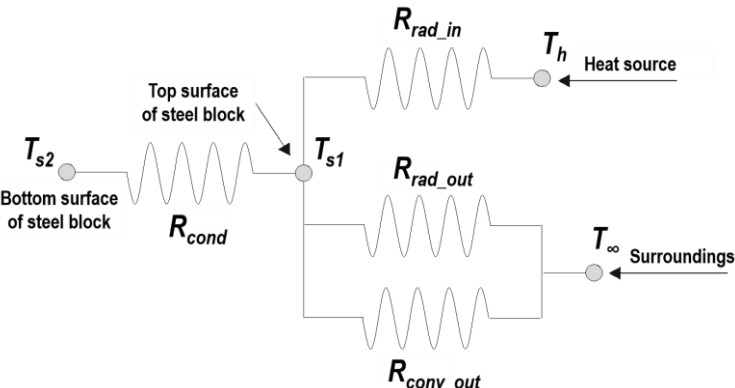

**Figure 3.** A diagram of thermal resistances in the heat transfer model using plain steel blocks.

### 2.3. Energy Balance

Under external and internal heat transmissions, the temperature of the lump of metal changes over time, as shown in Figure 4. The period in which the solid temperature rapidly increases is referred to as the transient period (or state) here. Its temperature increase rate decreases and approaches zero due to the continuous heat exchange between the exposed surfaces and the surrounding air. The period when the increase rate reaches zero is hereafter referred to as the steady state. To predict these time-dependent temperature developments in transient and steady states, a global conservation equation for energy was formulated as follows:

$$\rho_s c_{p,s} V_s \frac{dT_s}{dt} = \alpha_s \left( F_{h-top} \dot{Q}_h + F_{h-side} \dot{Q}_h^* \right) - \varepsilon_s \sigma \left( A_{top} + A_{side} \right) \left( T_s^4 - T_\infty^4 \right) \\ - \left( A_{top} h_{top} + A_{side} h_{side} \right) (T_s - T_\infty) \tag{5}$$

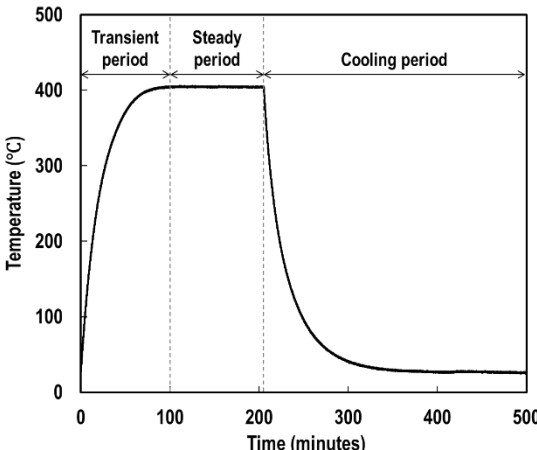

**Figure 4.** A typical time-temperature relationship of steel blocks tested with the cone calorimeter.

Equation (5) includes all the components of the thermal boundaries described in Figure 2. To achieve the prediction, the four critical parameters ($F$, $h$, $\varepsilon$, and $\alpha$) must be determined.

Followed by the process of temperature rise and thermal equilibrium, a subsequent profile of temperature can be plotted in a particular test environment in which only the phenomenon of heat loss occurs. This stage is referred to as the cooling period. This environment allows the mathematical terms

containing the absorptivity to be eliminated in the original governing equation, thus Equation (5) is modified as follows:

$$\rho_s c_p V_s \frac{dT_s}{dt} = \alpha_s \left( F_{top} \dot{Q}_h^0 + F_{side} \dot{Q}_h^{*0} \right) + \varepsilon_s \sigma \left( A_{top} + A_{side} \right) \left( T_s^4 - T_\infty^4 \right) \\ + \left( A_{top} h_{top} + A_{side} h_{side} \right) \left( T_s - T_\infty \right)$$

(6)

This modified version of energy balance can be used to determine the emissivity, which is discussed in Section 2.4.

## 2.4. Determination Procedure of Key Parameters

Amongst the key parameters, the view factor purely represents geometric configurations, and the convection coefficient is independent of the samples' material properties. These can therefore be individually derived from their specialised fields. Unlike these parameters, the emissivity and absorptivity are strongly case-dependent [23,24]. Their determination is significantly affected by a variety of attributes associated with the material and its surface condition (e.g., contamination, oxidation, temperature and roughness of surfaces; opaqueness; topology; and even chemical composition of the material). Particularly, the absorptivity is not truly a material property, also depending on the external radiation field relating to intensity, wavelength, polarization, and angle of incident radiation. Therefore, practical difficulties exist in defining the radiative properties in the field of engineering.

These radiative properties have long been a focus in the research areas where further detailed quantifications of radiant heat are needed (e.g., in industrial radiation thermometry, material optics, and microelectronic and laser material processing industries). Comprehensive studies in such fields have been conducted to determine either the emissivity or the absorptivity of specimens under various specified conditions (e.g., oxidation [23,24], surface roughness [24,25], wavelength [24,26–28], surface temperature [25–28], and chemical composition [25,27]). These existing data, however, are often sporadic and fluctuant, restricting use to only briefly estimating possible ranges of the properties for the given engineering-grade products. Due to the conventional use of Kirchhoff's approximation, most attention has focused on the emissivity, with less information available on absorptivity. From the viewpoint of the classic radiation principle, the determination of the absorptivity requires a number of spectrophotometer measurements under various conditions in relation to both the material and optical properties. A recent study [29] examined the effective absorptivity of several practical products exposed to different levels of irradiance. However, information on this spectral property of radiation also demonstrated the inherent complexities in its determination in the field of engineering.

An alternative approach to defining the radiative properties would be to allow numerical predictions of time-temperature profiles to be fitted to physical measurements for all stages from transient, through steady, and to cooling states, by regulating these properties. The best matched model verifies that the values employed are applicable for practical use. This numerical–experimental coupled method is valid only under particular conditions in which either the emissivity or the absorptivity is the last undefined parameter in the conservation equation for energy (Equation (5)). Hence, a three-phase procedure for determining the four controlling parameters was designed, as outlined in Figure 5.

In Phase I, the view factor and convective coefficient were individually determined. In Phase II, a possible range of the emissivity was identified from existing experimental data, once a certain degree of knowledge on the nature of the plain steel blocks and their surface condition has been achieved. The identified range is further narrowed using the numerical calculation method coupled with experimental data obtained during the cooling period. No radiation absorption mechanism occurs during this period. The emissivity is therefore the only undetermined parameter in Phase II. In Phase III, with the determined values of the three parameters (with the exception of the absorptivity), a possible range of the last undefined parameter (i.e., $\alpha$) was identified in the same manner as shown

in Phase II. The time–temperature profiles measured in the transient and steady states were used in the numerical–experimental coupled method.

Notably, the value of $\alpha$ determined in the third phase is not strictly a genuine absorptivity value based on the classic principle of electromagnetic wave theory. Since the parameter was defined after completing the determination of $F$, $h$, and $\varepsilon$ in Phases I and II, it primarily represents the phenomenon of radiation absorption and secondarily the experimental uncertainties. The determined value was hence appointed the role of the effective factor.

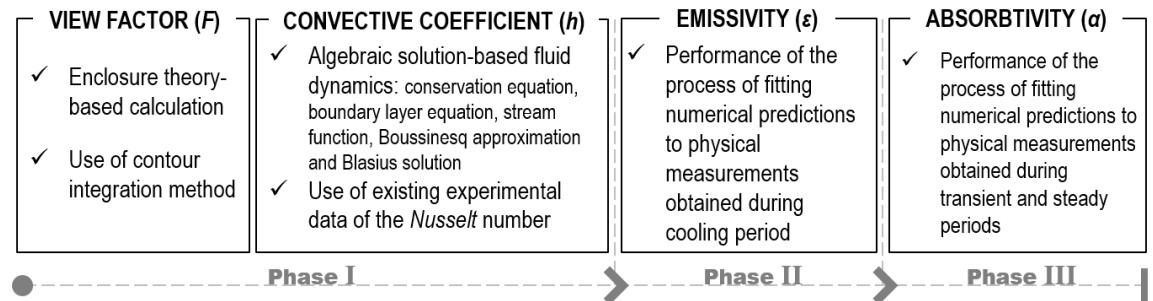

**Figure 5.** Determination procedure of controlling parameters representing thermal boundaries.

With respect to this approach, in prior studies, a similar technique was used to assign the effective role to the convection coefficient $h$ [4,18–20]. This assignment implies that $h$ represents convection transfers, in addition to all the remaining uncertainties regarding radiation transfers and experiments. This is because specific determinations of the other critical parameters (i.e., $F$, $\varepsilon$, and $\alpha$) were not adequately established in advance. In contrast, the effective factor used here was isolated from the determinations of the other key parameters (i.e., $F$, $h$, and $\varepsilon$) and only included experimental uncertainties. This approach is based on the proposition that the radiation-related parameters, $F$, $\varepsilon$, and $\alpha$, are too influential to be represented solely by the convection-related parameter $h$ under the high temperature conditions created during cone calorimetry tests.

## 3. Experimental Details

A series of cone calorimetry tests were conducted using the plain steel blocks to verify numerical predictions of the thermal boundaries. Low-carbon steel with a carbon content of less than 0.28% is frequently used for welded structures. This material element was manufactured in sets of blocks, $100 \times 100$ mm$^2$ with thicknesses ($\delta$) of 10, 15, and 20 mm, as illustrated in Figure 6a. To measure the time-dependent temperature development of the lump of the metals, four k-type thermocouples were welded to its bottom surface; this placement was used to ensure that the sensors were not directly exposed to irradiance, and multiple gauging was used to evaluate the credibility of the measurements. The bottom surface was then insulated by low density materials, as shown in Figure 6b. No specimen holder was used. All the exposed surfaces were smoothed to minimise uncertainties caused by any contamination or oxidation but were not polished using either grinders or chemical liquids. With respect to the surface condition, once a specimen was tested, its reuse was prohibited, as its surface underwent oxidization during the testing process.

Figure 7 demonstrates the various test configurations in accordance with the use of the specimens with different thicknesses. The distance between the baseplate of the frustum of the heater and the top surface of the 10-mm-thick block was initially adjusted to 25 mm, which is a typical setup adopted in fire safety engineering. The distance was then designed to be altered according to the thickness of the samples placed in the apparatus to examine the change in the quantity of the net heat store in relation to the phenomenon of volume expansion of the given samples during testing. The amount of the irradiance, physically measured at the centre along the cone axis, 25 mm underneath the heater's baseplate in the calibration stage ($\dot{q}''_{irr}$), was regulated to 35, 50, and 65 kW/m$^2$. This heat flux setting is the standard level of external thermal loading that enables the chemical degradation

of bench-scale samples to be activated and observed. The centrifugal exhaust fan was turned off during the tests to minimise possibilities of the generation of forced fluid motions around the specimen tested. With respect to the procedure of the cooling test, the sample was fully heated up to a steady temperature and spatially isolated from the heater. It thus continuously lost part of the stored heat to its surroundings at ambient temperatures via natural convection and radiative emission. The cooling part of the entire time–temperature profile, which was introduced in Figure 4, was obtained from this test. The test for each of the specimens with different thicknesses under different heating conditions was triplicated to ascertain the repeatability of the physical measurements.

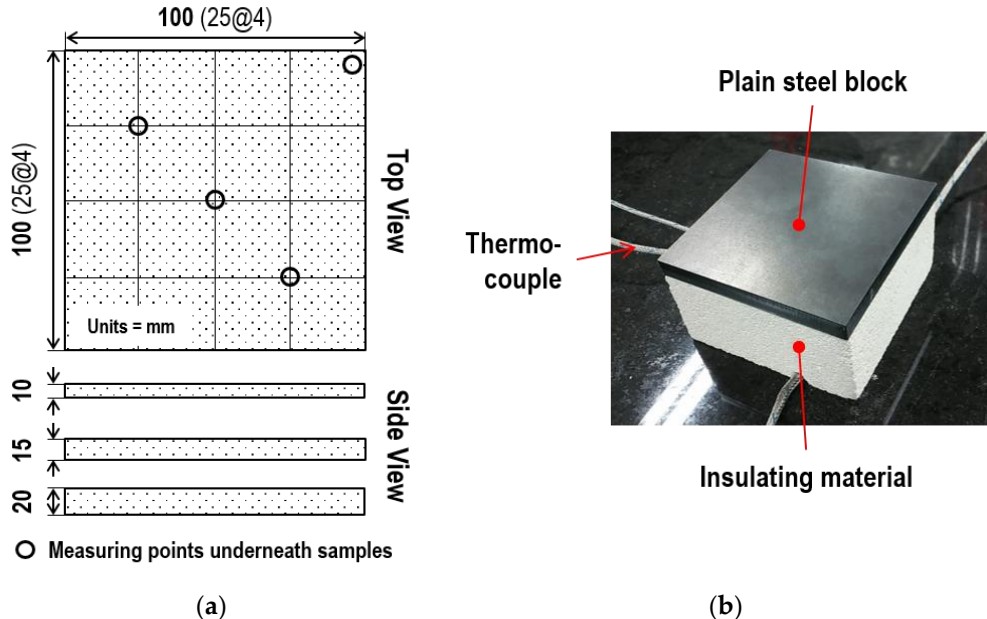

(**a**) (**b**)

**Figure 6.** Specimen dimensions (**a**) and photographic view of specimen setup (**b**).

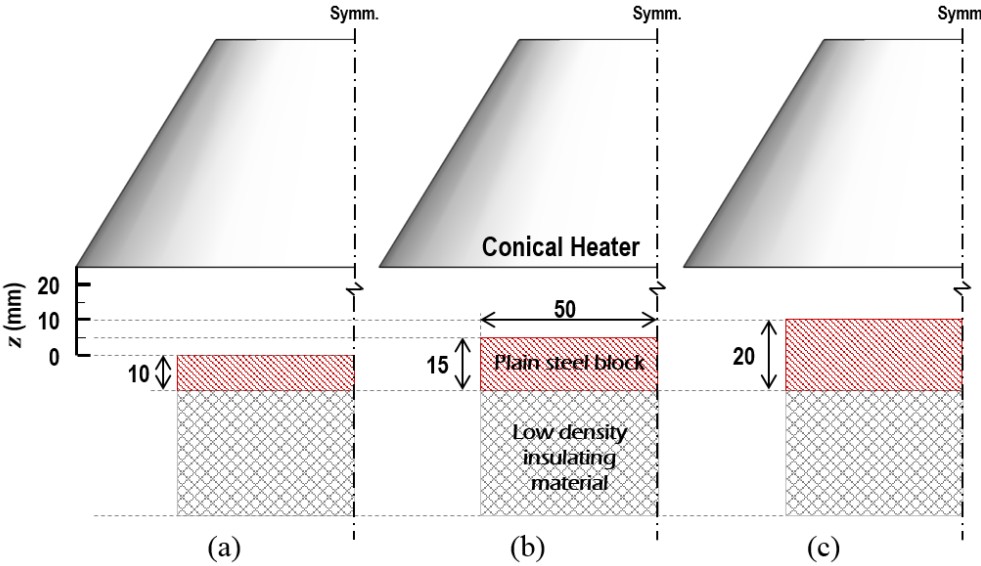

(a) (b) (c)

**Figure 7.** Test configurations in cone calorimetry tests for specimens with thicknesses ($\delta$) of (**a**) 10, (**b**) 15, and (**c**) 20 mm.

## 4. Determination of Key Parameters

The controlling parameters were defined in compliance with the determination procedure shown in Figure 5. Amongst the four parameters, the view factor was previously determined in Section 2.1.2 and is listed in Table 1.

### 4.1. Convection Heat Transfer Coefficient

Buoyancy-induced fluid motion has long been the focus of research in the field of heat and mass transfer. It is a common practice to characterise these physical phenomena using a typical dimensionless form of the Nusselt number ($Nu = C(Ra)^n$) based on algebraic solutions [30,31]. This theoretical characterization establishes a notional link between fluid motion and convective heat transfer, both occurring at solid-fluid interfaces. It is therefore conventional to assess $Nu$ in defining convection transfers represented by the coefficient $h$. The determination of $Nu$ is significantly affected by three factors: (1) heating conditions, (2) adjacent surface orientation (in terms of the gravitational field), and (3) flow regime. To define the correlations of $Nu$ appropriate for the flows anticipated during cone calorimetry tests, characterizing the fluid motion is a prerequisite, and is based on three factors:

- In the tests, specimens' exposed surfaces were heated consistently by a uniform radiant flux density emitted by the conical heater. Prior research [32,33] examined free convection from a vertical plane under such heating conditions, differentiated from the conventional condition for uniform wall temperatures, and developed a modified Rayleigh number. However, the deviations between the correlations obtained under the two different heating conditions are negligible in the case of free convective flow, adjacent to the vertical and horizontal planes [34]. Hence, the conventional version of the Rayleigh number, which has been more widely examined, was adopted for the steel block model studied in this work.

- Regarding surface orientation, the use of gravitational term $g \cos \theta$ in place of the gravitational acceleration ($g$ in m/s$^2$) in the typical Grashof number ($Gr$) was suggested for inclined surfaces, including horizontal surfaces [35], where $\theta$ is the angle of inclination from vertical in radians. This approach was capable of correlating well with the classical studies on free convective motion adjacent to vertical planes [36]. However, the correlations were not satisfactory for the horizontal top surface of the heated blocks [37], and even the turbulence data in this orientation from the air provide a closer correlation to the general $Gr$ than with the modified $Gr$ [34]. Considering these assertions, the conventional $Gr$ was adopted for the convection on the top surface of solids in this work.

- With respect to the flow regime, the onset of the transition in free convection over vertical planes occurs when $Ra$ is in the order of 2h10$^8$ [38]. For inclined, upward-facing planes under uniform heat fluxes, a correlation for the critical $Ra$ was derived to be $6.31e10^{12}e^{-11\theta}$ [39]. According to these suggestions, the patterns of fluid motions anticipated in this instrument were evaluated as laminar and turbulent for the side and top surfaces of the heated steel bodies, respectively.

Considering these attributes of convection in the cone calorimeter, the free convections generated on the two differently oriented exposed surfaces can be specified into: (1) laminar free convection adjacent to a vertical surface with uniform heating or (2) turbulent free convection adjacent to a horizontal upward-facing surface with a uniform wall temperature.

To determine $h$, applicable ranges of $Nu$ were deduced from existing experimental data and their nonlinear regression models. Theoretical assessments of $Nu$ tend to be dependent of empirical data due to the complexity of fluid motions, but conducting a test on the convective motions over horizontal and vertical planes, which are prevalent in the field of fluid dynamics, is beyond the scope of this work.

Figure 8a shows the measurement data points for free convection flows on vertical planes under uniform heating [34], with $Nu$–$Ra$ relationships predicted using Equations (7)–(9) originating from prior works [33,40].

$$Nu = 0.67(Raf(\text{Pr}))^{1/4} \tag{7}$$

$$Nu = \left\{0.825 + 0.387(Ra\psi(\mathrm{Pr}))^{1/6}\right\}^{2} \tag{8}$$

$$Nu = 0.68 + 0.67(Ra\psi(\mathrm{Pr}))^{1/4} \tag{9}$$

where $f(Pr) = \left[1 + \left(\frac{0.437}{Pr}\right)^{9/16}\right]^{-16/9}$ and $\psi(Pr) = \left[1 + \left(\frac{0.492}{Pr}\right)^{9/16}\right]^{-16/9}$. The empirical regression models enabled the corresponding value of the dependent variable on the ordinate $\log(Nu)$ to be predicted if the value of the independent variable on the abscissa, $\log(Ra\varnothing(Pr))$, could be estimated. In the estimations, the value for $\log(Ra\varnothing(Pr))$ varied according to the temperature increase of the metals, primarily due to the dependence of $Gr$ on the temperature difference between the steel bodies and their surroundings (i.e., $T_s - T_\infty$), and secondarily to the temperature-dependent thermophysical properties of air. This variation was limited to between approximately 1.94 and 4.19, as expressed by the shaded grey area in Figure 8a. This variation occurs in relation to the prespecified heat flux setting. A range of the $\log(Nu)$ value was subsequently estimated using Equation (8), which drew the best-fit regression line to the overall experimental data. This resulted in upper and lower bounds of 0.88 and 0.43, respectively. Considering the correlation between $Nu$ and $h$, we concluded that the value of $h_{side}$ (for convective heat transfer on the side surfaces of specimens) varied between approximately 5.14 and 14.51 W/m$^2$K with the surface temperature between approximately 20 and 563 °C.

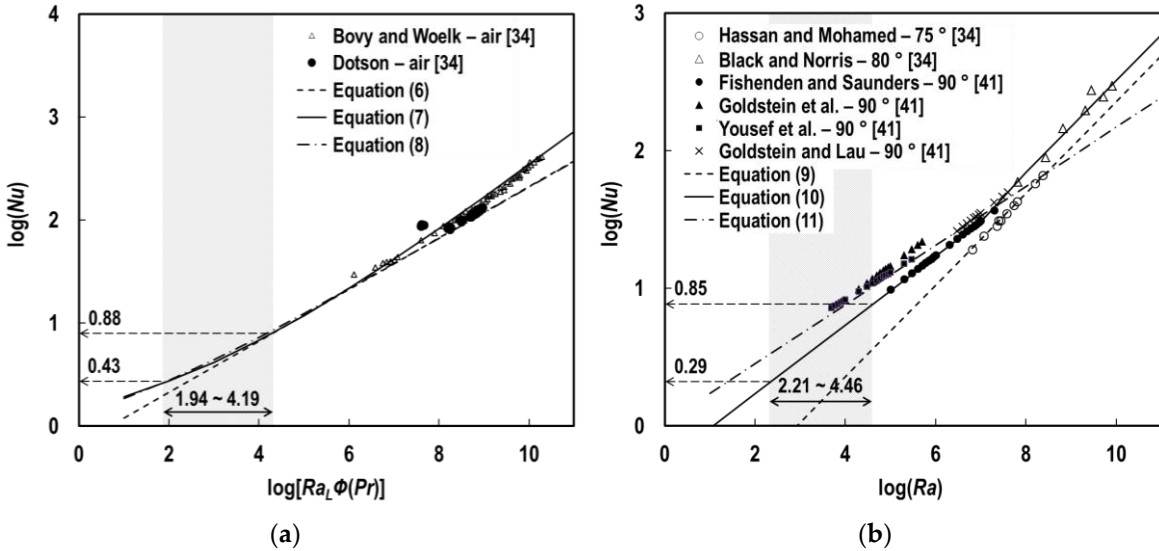

(a)  (b)

**Figure 8.** *Ra–Nu* relationships for free convection from uniformly heated, (**a**) vertical, and (**b**) inclined and horizontal plates, to air.

The other coefficient value, $h_{top}$, was determined following the same method in the previous derivation of $h_{side}$. Figure 8b shows the scatter marked with symbols for large angles of inclination, which was measured in prior studies [34,41] and its regression lines were estimated by

$$Nu = 0.15(Ra\psi(\mathrm{Pr}))^{1/3} \tag{10}$$

$$\begin{aligned} Nu &= 0.54Ra^{1/4} \ (Ra \leq 10^7), \ \mathrm{and} \\ Nu &= 0.15Ra^{1/3} \ (Ra > 10^7) \end{aligned} \tag{11}$$

$$Nu = 1.05Ra^{0.215} \tag{12}$$

Equation (11) yielded a regression model in better agreement with the existing overall data, which was applied in predicting a range of the value of the independent variable on the abscissa, $\log(Ra)$. Its range was estimated as 2.21–4.46, resulting in upper and lower limits of the corresponding value of the dependent variable on the ordinate, $\log(Nu)$, of 0.85 and 0.29, respectively. Under the

circumstances, $h_{top}$ remained in the range of approximately 4.25 to 15.48 W/m²K during the transient and steady periods.

Table 2 shows the coefficient values used in prior analytical studies [3,4,7,8,16–22] that assessed the reliability of the values predicted in this work; most of the data are for $h_{top}$, whereas the values with the superscript "a" are from tests used to examine the effect of vertical burning with samples rotated through 90°. The ranges of the coefficient values determined in this work overlapped with those having been commonly used in fire safety engineering. Consequently, the ranges of the coefficients $h_{top}$ and $h_{side}$ averaged to 14.57 and 13.67 W/m²K, respectively, over the entire heating period. These values were used in subsequent calculations of the global conservation equation for energy.

**Table 2.** Convective coefficients used in previous studies [3,4,7,8,16–22].

| Author | Convection Heat Transfer Coefficient (W/m²K) |
|---|---|
| Janssens (1991) | 9–27 [a] |
| Rhodes et al. (1996) | 10 |
| Hopkins Jr. et al. (1996) | 10 |
| de Ris et al. (2000) | 7.6 |
| Janssens (2002) | 3.9–17.1 [a] |
| Bartholmai et al. (2003) | 20 |
| Wang (2005) | 20 |
| Lautenberger et al. (2006) | 10 |
| Staggs (2009) | 15.15–25.25 |
| Zhang et al. (2009) | 7–15 |
| Mesquita et al. (2009) | 20 |

### 4.2. Emissivity

For the identification of a possible range of emissivity for the plain steel objects, we started by reviewing prior works from the viewpoints of chemical composition and surface condition. According to the Korean Industrial Standards and manufacturer's products guide [42,43], the low carbon steel used in the tests (SM400) typically contains less than 0.28 wt.% carbon (C), 1.60 wt.% manganese (Mn), 0.035 wt.% phosphorus (P), and 0.035 wt.% sulphur (S). A prior study [44], which intensively examined thermal radiative properties for a variety of metals, provided experimental information for low carbon steel, which contained 0.18–0.23 wt.% C, 0.30–0.60 wt.% Mn, and less than 0.05 wt.% P and 0.04 wt.% S. Based on the data, a possible range of the emissivity to the plain steel blocks was identified as 0.830–0.945.

This identified range was further narrowed using the numerical–experimental coupled method with Equation (6) and the data physically measured from the cooling tests. In the calculation of the energy balance equation using FDM, the defined parameters in Phases I and II ($F_{h\text{-}top}$ and $F_{h\text{-}side}$ listed in Table 1, and $h_{top}$ = 14.57 W/m²K and $h_{side}$ = 13.67 W/m²K) and nonlinear temperature-dependent material properties of carbon steel and air [37,45] were used. To demonstrate the degree of agreement between the predictions and measurements, a discrete form of standard error ($\omega$) was calculated using

$$\omega = \sqrt{\frac{\sum_{i=1}^{N}\left[T_{\exp}(i) - T_{cal}(i)\right]^2 \Delta t}{t_{total}}} \tag{13}$$

where $t_{total}$ refers to the total time (s) of experiments.

Figure 9a,b demonstrates the variations in the standard error in regulating the emissivity within the initially limited range of 0.830 to 0.945 at a constant irradiance of 50 kW/m² or a constant thickness of 15 mm, respectively. The minimum standard error was obtained in a range of $\varepsilon$ between 0.87 and 0.90. Figure 10a,b shows the comparative outcomes of the cooling profiles of the samples with different thicknesses under different heating conditions. Each dotted line with colours indicates the experimental data gauged by a single thermocouple, whereas the black lines show the calculations at $\varepsilon$ = 0.88.

We concluded that the value of 0.88 yielded the predictions in good agreement with the measurements. Hence, the emissivity was determined as this value, which was used in subsequent calculations of the governing equation.

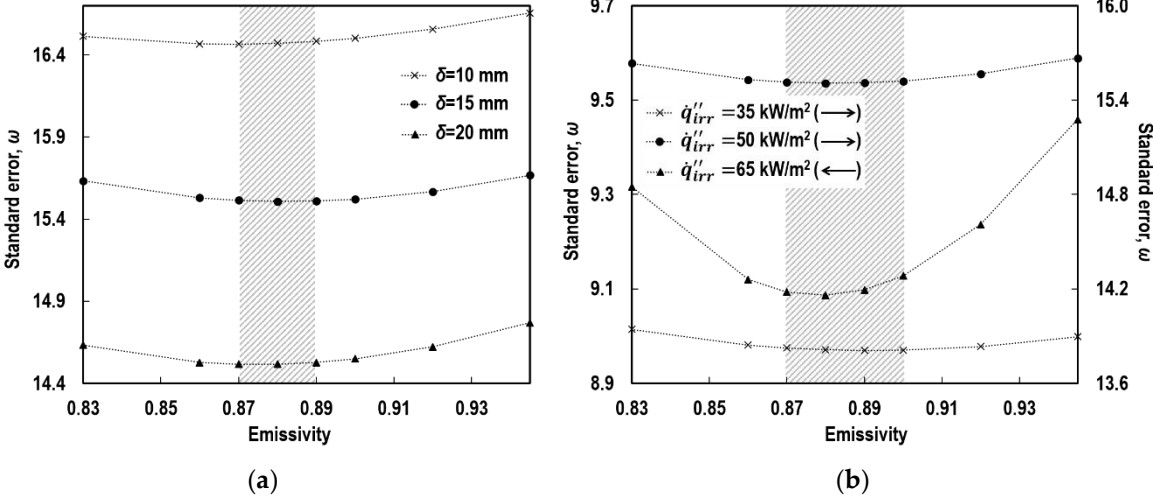

**Figure 9.** Variations in standard error according to the alteration of emissivity at (**a**) $\dot{q}_{irr}'' = 50$ kW/m$^2$ and (**b**) $\delta = 15$ mm.

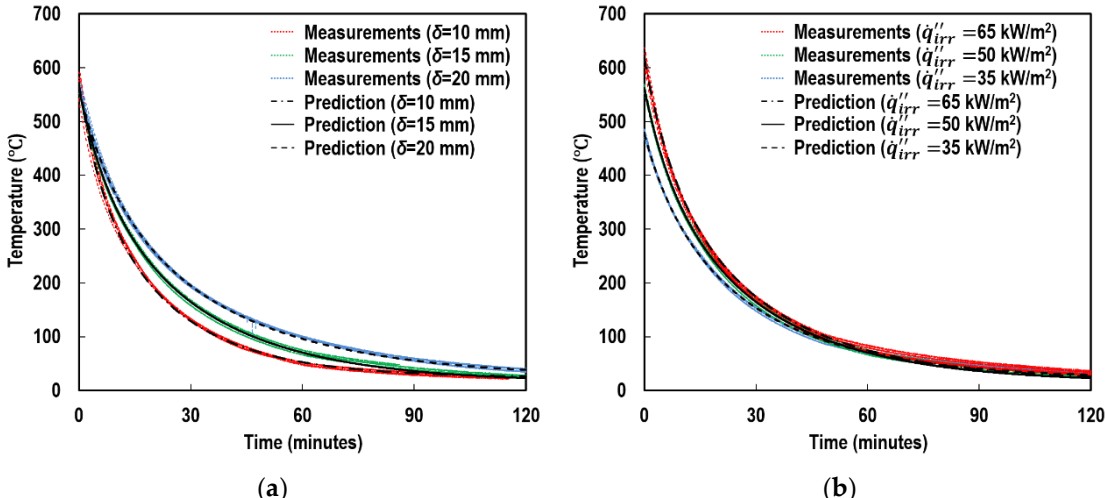

**Figure 10.** Time–temperature profiles during cooling periods at (**a**) $\dot{q}_{irr}'' = 50$ kW/m$^2$ and (**b**) $\delta = 15$ mm.

### 4.3. Absorptivity

The inherent difficulties in defining the absorptivity conversely make this property a good candidate for use as an effective factor in numerical predictions of specimens' temperature developments. The overall determination process of the applicable range of $\alpha$ to the steel block model is analogous to that of the emissivity discussed in Section 4.2. Figure 11a,b exhibits the variations in the standard error according to the alteration of the absorptivity. The minimum values of $\omega$ for the blocks with three different thicknesses at the three different irradiances stayed in a range of $\alpha$ between 0.76 and 0.80, as indicated by the shaded grey regions.

Figure 12a,b illustrates the time–temperature profiles of the plain steel blocks in transient and steady states obtained from both the experiments and the numerical calculations. The coloured dotted lines refer to the data sets of measurements, whereas the black lines superimposed on the data show their corresponding predictions at $\alpha = 0.78$. Regarding the transient state, we identified that all the estimations of the rate of temperature rise (i.e., d$T_s$/d$t$) were in good agreement with the test results,

regardless of the alterations of both the samples' thicknesses and heating conditions. This indicates that the proposed model accurately quantifies the net heat stored in every differential time interval during the transient periods. With respect to the steady state, we found that there were little discrepancies between the steady temperatures numerically predicted and physically measured.

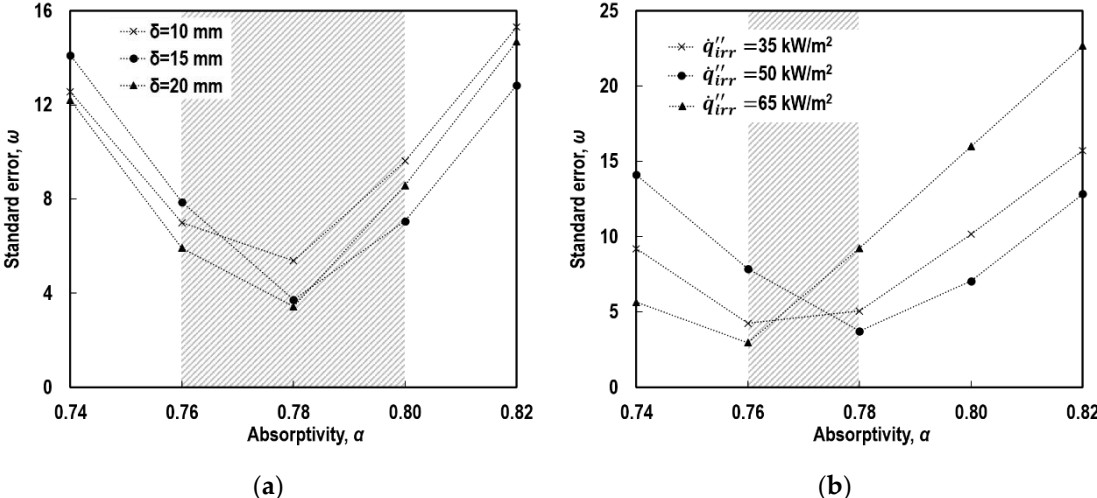

**Figure 11.** Variations in standard error according to the alteration of absorptivity at (**a**) $\dot{q}''_{irr}$ = 50 kW/m² and (**b**) δ = 15 mm.

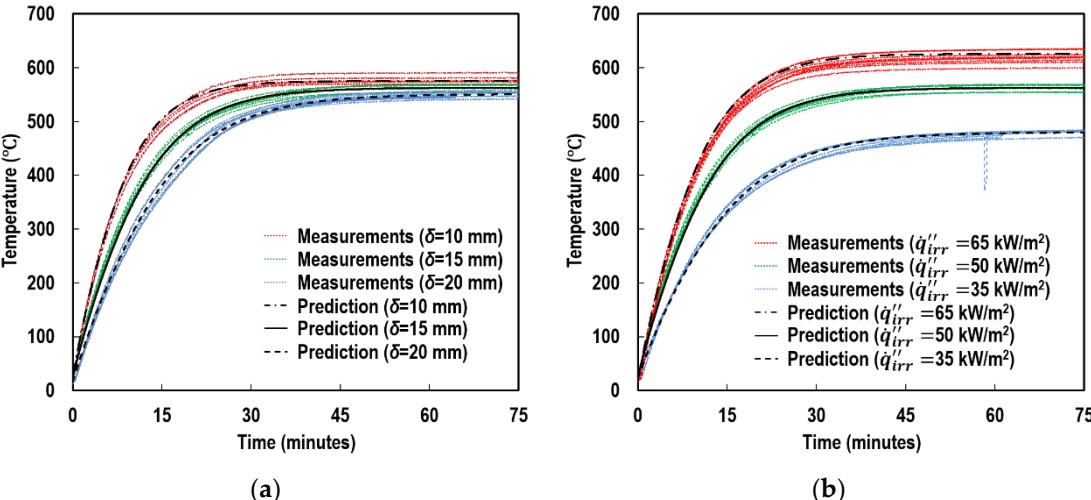

**Figure 12.** Time–temperature profiles during transient and steady periods at (**a**) $\dot{q}''_{irr}$ = 50 kW/m² and (**b**) δ = 15 mm.

## 5. Discussion

The time-dependent temperature development of the plain steel block is determined by its thermophysical properties and the combination of the multiple heat transfers generated on its exposed boundaries. The components previously specified with the symbols of Ⓐ–④ in Figure 2 were separately quantified to identify their individual impacts. These quantities were then divided by the total amount of the heat gained and lost, and these ratios were converted to percentage terms to facilitate comparison. The terms were categorised into (1) heat gain/loss, (2) on top/side surfaces, and (3) via radiation/convection modes, which are demonstrated in Figure 13a,b. In the bar charts, the heat gain and loss are expressed by two independent bars (i.e., I and II). The heat transfers occurring on the top and side surfaces are indicated by solid and hatched patterns, whereas radiation and convection modes

are depicted in black and red, respectively. The heat transmitted during the transient and steady states was also independently analysed.

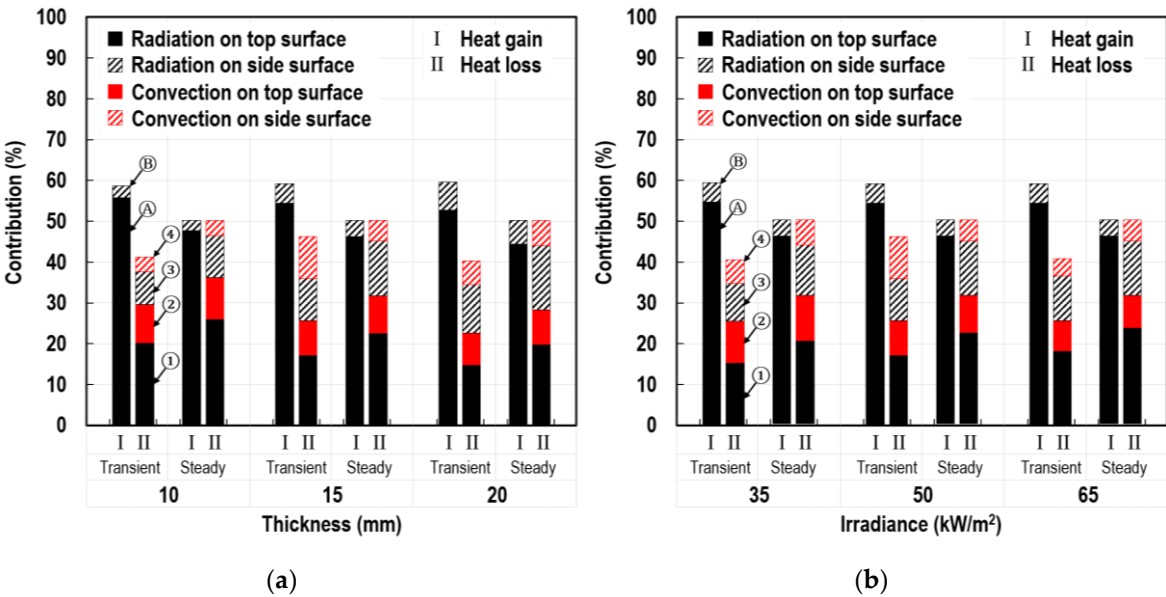

**Figure 13.** Individual contributions of heat transfer mechanisms during transient and steady periods at (**a**) $\dot{q}_{irr}'' = 50$ kW/m$^2$ and (**b**) $\delta = 15$ mm.

Unlike under steady states, each portion of the distribution varies with time during transient periods. To comparatively analyse these time-dependent proportions, the amounts of heat corresponding to each symbol (Ⓐ–④) were mathematically integrated over the overall transient period. The inflection point, at which a transition from transient to steady states occurs, was determined at the instant when the rate of temperature rise ($dT/dt$) became less than 0.01. This criterion resulted in approximately 37.9, 48.4, and 56.8 min for the solids with thicknesses of 10, 15, and 20 mm at a constant irradiance of 50 kW/m$^2$, respectively, and 55.6 min and 43.4 min with a constant thickness of 15 mm at the irradiances of 35 and 65 kW/m$^2$, respectively.

### 5.1. Individual Contributions of Top and Side Surfaces

The heat transmitted through the top surface comprised a larger proportion of the total incoming and outgoing energies than that by the side surfaces. The amount of heat transferred on the top surface (Ⓐ, ①, and ②) accounted for between approximately 72.2% and 85.4% of the total energy transmitted (Ⓐ–④).

In specific regard to the heat gain mechanism, the top surface (Ⓐ) was still the dominant area through which most of the heat released by the heater was absorbed. A gradual growth in the contribution of the side surfaces (Ⓑ) was observed by increasing the thickness of the solids. At $\delta = 20$ mm, the amount of heat gained by the sides accounted for approximately 11.6% of the total incoming heat (Ⓐ and Ⓑ), and its proportion could increase to 16.5% if the thickness was 30 mm.

The contribution of the side surfaces to the heat loss process (③ and ④) was more appreciable than that to the heat gain mechanism (Ⓑ). The contribution became significant as the thickness of samples increased, resulting in up to approximately 44.0% of the total heat loss (①–④). This is in direct relation to the surface area involved; the side surface area (8000 mm$^2$) is close to the top surface area (10,000 mm$^2$) at $\delta = 20$ mm. These findings indicate that the exposed perimeter surfaces have a considerable effect on the quantification of the heat loss process.

### 5.2. Individual Contributions of Radiation and Convection Modes

Radiation was the dominant heat transfer mode in cone calorimetry tests. The radiation mode (Ⓐ, Ⓑ, ①, and ③) comprised up to approximately 88.0% of the entire heat exchanged (Ⓐ–④). In specific terms of the heat loss mechanism, the radiant emission (① and ③) accounted for between 60.6% and 73.9% of the total heat loss (①–④). These are related to the finding that the blocks were heated up to relatively high temperatures in the tests (480–626 °C).

Despite the dominance of radiation, a considerable proportion was due to convection (② and ④), which was in the range of approximately 26.1% to 39.4% of the total heat loss. Its minimum contribution was observed at the highest level of irradiance used in the tests (65 kW/m$^2$). This indicates that under severe heating conditions, the contribution of convection loss becomes less noticeable.

The contribution of convective heat lost from the side surfaces became comparable to that from the top surface as the thickness of the blocks increased. This is in direct relation to the increase in the exposed perimeter surface area. This comparable portion regarding convection over the side surfaces indicates the importance of defining the convective coefficient for the vertically oriented surfaces, differentiated from that of the typical horizontal surface.

### 5.3. Individual Contributions of Radiant Absorption and Emission

The findings about the dominance of radiation during testing suggest, the need for an advanced understanding of the individual influences of the radiative properties, $\varepsilon$ and $\alpha$, on the overall heat transmission. In transient states (when the amount of radiant absorption is greater than that of radiant emission), the absorptivity was incorporated into the mechanism of radiant absorption (Ⓐ and Ⓑ) which accounted for up to 59.6% of the total energy exchanged (Ⓐ–④). This portion was approximately 2.2 times greater than that with regard to the mechanism of radiant emission, which was represented by the emissivity. Even when the two types of radiation transfer reached thermal equilibrium, we found that the quantity of radiant absorption was more than 1.4 times larger than that of the radiant emission.

These data sets demonstrate the influence of absorptivity in comparison to emissivity during cone calorimetry testing, suggesting the need to develop an improved approach over both the conventional approximation (which assumes the absorptivity being equivalent to the emissivity) and prior approaches (which assigned the role of the effective factor to the convection-related parameter $h$). This work, therefore, contributes to a better understanding of the temperature increase of specimens tested with this apparatus, particularly during transient periods.

## 6. Conclusions

In this study, we examined the uncertainties surrounding the technical aspects of specimens' thermal boundaries. We specifically investigated block-shaped specimens in a cone calorimeter. In this case, the use of the conventional, one-dimensional heat transfer model led to apparent discrepancies in predicting the amount of net heat stored in differential time intervals, particularly during transient periods. These errors would have nullified the assessments of the flammability or thermal performance of the test samples. This paper presented a process to determine the four critical parameters representing the thermal boundaries: view factor, convection heat transfer coefficient, emissivity, and absorptivity. These were substantiated by comparison with a series of cone calorimetry tests using plain steel blocks. With the defined parameters, the individual contributions of each of the specified heat transmissions simultaneously occurring during the tests were quantitatively analysed.

We found that up to 59.6% of the total energy exchanged was accounted for by the radiant absorption mechanism, and the quantity was significantly affected by the view factor and the absorptivity. Hence, these parameters must be carefully considered when predicting the net heat absorbed. As a result of the independent consideration of the absorptivity, being differentiated from the emissivity, good agreement on specimens' time–temperature relationships was observed between the numerical predictions and experimental measurements for all stages, from transient, through steady,

and to cooling states. This could not have been achieved if the conventional Kirchhoff's approximation was used. When specifically examining radiation transfers, through the use of percentage distributions, we demonstrated that the influence of absorptivity was greater than that of emissivity, becoming particularly evident during transient periods.

The pure geometric view factor between the truncated, cone-shaped emitter and recipient, indicating the horizontal top or vertical perimeter plane, was calculated using the contour integration method. The fluid dynamic coefficient of convection was derived from existing data related to fluid motion and its regression model. The assessment process of these parameters was evident, based on the theories in their specialised fields (i.e., enclosure theory and fluid mechanics, respectively). Therefore, the results obtained can be generalised for practical applications. However, we recognised that the determination of the radiative properties, emissivity and absorptivity, was a more complex process due to their dependence on a variety of attributes of the materials and heat sources used. The evaluation method proposed in this work contributes to the ability to identify these properties for engineering grade materials in the field of fire safety.

With respect to exposed surface area, we demonstrated that the majority of incoming and outgoing heat is transmitted through the top surface of specimens via radiation. However, we concluded that the heat transmitted through the perimeter side surfaces, including in the convection mode, still needs to be carefully considered, particularly in the examination of the heat loss mechanism. The amount of heat lost is proportional to the quantity of surface area exposed to surrounding air, whereas this quantity, as well as the view factor, should be considered together when quantifying the heat gain mechanism.

Using this determination process and considering the evaluation of the findings with regard to each individual contribution, advanced strategies for managing variations in the thermal boundaries during cone calorimetry testing can be established.

**Author Contributions:** Conceptualization, S.K. and S.C.; Data Curation, J.Y.C.; Formal Analysis, S.K.; Funding Acquisition, J.Y.C. and S.C.; Investigation, M.K.; Methodology, S.C.; Project Administration, M.K.; Resources, M.K.; Software, S.K. and S.C.; Writing—Original Draft, S.K. and S.C.; Writing—Review and Editing, J.Y.C.

**Funding:** This research was funded by the National Fire Agency of the Republic of Korea through the R&D program both on (1) the Development of an Emergency Response Training Facility for Alternative Vehicles Related to Fire, and (2) No. 2018-NFA002-004-01010000-2018.

**Conflicts of Interest:** The authors declare no conflict of interest.

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
