# Peer review of "Thermal Boundaries in Cone Calorimetry Testing"

_coatings, doi:10.3390/coatings9100629_

Round 1

Reviewer 1 Report

This manuscript addresses the difficulties of evaluation of materials which intumesce during cone calorimetry. This is an important area of concern and the results presented offer a better definition of the issues to be considered when testing samples of this kind. The manuscript will benefit from some revision for clarity and readability. Corrections are penciled-in directly on pages of the manuscript attached.These are indicative of the kinds of changes needed throughout. Such statements as "literature did not take into consideration" are incorrect and do not state what was intended. It isn't the literature that considered (literature is not active) but rather previously reported studies. Similarly, "existing literature" should be "previous work"; "this paper examined" should be "the work described in this paper examined"; "mechanism" is loosely used and should generally be "process". Proper tenses should be used throughout.

Author Response

The expressions that the reviewer pointed out were revised. On top of that, we will request English editing service of MDPI to improve the quality of our manuscript.

Reviewer 2 Report

Dear Authors,

a modelisation or at least a discussion of a real polymeric thick coating seems necessary to increase the interest of the future readers.

What are the limits of your model according polymeric coating (porosity, emmissivity) ?

Can you estimate the proportion of energy exchanged by the different mechanisms for such polymeric coatings ?

Etc.

Author Response

(Q1) A discussion of a real polymeric thick coating seems necessary to increase the interest of the future readers.

This manuscript has a series of companion papers as the research project was conducted in sequence to overcome difficulties in analysing a complex interacting thermo-physical behaviour of an inorganic intumescent coating: (1) experimental study of thermo-kinetic decomposition reactions of the polymeric coating; (2) clarification of thermal boundaries of a ‘thick’ block-shaped specimen in cone calorimeter testing; (3) numerical analysis of the internal heat transfer through a thick porous medium (i.e. fully expanded coating) in cone calorimeter testing; (4) numerical and experimental study of the integrated dynamic behaviour of the coating (i.e. intumescence). Three of the topics were already discussed in published papers [11-13], respectively. This manuscript intensively discusses the second and does not directly deal with the polymeric material. Therefore, instead of addressing a discussion about the real polymer so that this may make readers get confused, we will add information of the companion papers in the section of Introduction to help readers’ understanding.

Topic (1) - [11] Kang, S.; Choi, S.; Choi, J.Y. Numerical prediction on interacting thermal-structural behaviour of inorganic intumescent coating. In: International Conference and Exhibition on Fire Science and Engineering (Interflam), Nr Windsor, UK, 4-6 July 2016, pp.213-224.

Topic (3) - [12] Kang, S.; Choi, S.; Choi, J.Y. Mechanism of heat transfer through porous media of inorganic intumescent coating in cone calorimeter testing. Polymers, 2019, 11, 221.

Topic (4) - [13] Kang, S.; Choi, S.; Choi, J.Y. Coupled thermos-physical behaviour of an inorganic intumescent system in cone calorimeter testing. J. Fire Sci., 2017, 35(3), 207–234.

(Q2) What are the limits of your model according polymeric coating?

The model and approach addressed in this manuscript can be generally used for any block-shaped samples placed in the cone calorimeter (which was discussed in the section of Conclusions), whilst the model according polymeric coating is not discussed in this paper but in companion papers [12-13]; the latter model excludes the degradation phase of the coating after being fully expanded because the inorganic material studied in this project does maintain its fully expanded status within the thermal environment created by the cone calorimetry once inflated.

(Q3) Can you estimate the proportion of energy exchanged by the different mechanisms for such polymeric coatings?

It is fairly challenging to quantify the respective proportions of energy-in and -out through each of the top and perimeter side surfaces in different heat transfer modes, particularly for such polymers with dynamic behaviours. This is because (1) both the ‘external’ heat exchange (upon the exposed surfaces of the specimen) and ‘internal’ heat transmission (through the porous medium) play key roles on the determination of the temperature increase and temperature distribution as a function of time; (2) the temperature changes of the polymeric material lead to the changes in its thermo-kinetic reactions, its geometric features (such as porosity and thickness), thermal exposure condition (due to the stationary conical heater), etc. in real time. Under the circumstances, we simplified the composite process to make clearer; we examined the external thermal boundaries and proportion (or contribution) of energy exchanges in this manuscript, using a plain steel block to simplify the internal heat transmission (i.e. lumped capacitance approach); We studied the internal heat transmission and contributions of heat transfer modes in the prior work [12].